# Field Research on Mixing Aeration in a Drinking Water Reservoir: Performance and Microbial Community Structure

**DOI:** 10.3390/ijerph16214221

**Published:** 2019-10-31

**Authors:** Zizhen Zhou, Tinlin Huang, Weijin Gong, Yang Li, Yue Liu, Shilei Zhou

**Affiliations:** 1School of energy and environment, Zhongyuan University of Technology, Zhengzhou 450007, China; 6623@zut.edu.cn (W.G.); ly_zut@163.com (Y.L.); yue5757@sina.com (Y.L.); 2School of environmental and municipal engineering, Xi’an University of Architecture and Technology, Xi’an 710055, China; 3Key Laboratory of Northwest Water Resource, Environment and Ecology, MOE, Xi’an University of Architecture and Technology, Xi’an 710055, China; 4School of environment science and engineering, Hebei University of Science and Technology, Shijiazhuang 050018, China; zslzhoushilei@126.com

**Keywords:** drinking water reservoir, water-lifting aeration, microbial community, environmental factors

## Abstract

Field research on the performance of pollutant removal and the structure of the microbial community was carried out on a drinking water reservoir. After one month of operation of a water-lifting aeration system, the water temperature difference between the bottom and the surface decreased from 9.9 to 3.1 °C, and the concentration of the dissolved oxygen (DO) in the bottom layer increased from 0 to 4.2 mg/L. The existing stratification in the reservoir was successfully eliminated. Total nitrogen (TN), total phosphorus (TP), and total organic carbon (TOC) concentrations were reduced by 47.8%, 66.7%, and 22.9%, respectively. High-throughput sequencing showed that *Proteobacteria*, *Bacteroides,* and *Actinomycetes* accounted for 67.52% to 78.74% of the total bacterial population. Differences in the bacterial changes were observed between the enhanced area and the control area. With the operation of the water-lifting aeration system, the populations of bacteria of the main genera varied temporally and spatially. Principal component analysis pointed out a clear evolution in the vertical distribution of the microbial structure controlled by the operation of the aeration system. Permutational analysis of variance showed a significant difference in the microbial community (*p* < 0.01). Redundancy analysis showed that physical (water temperature, DO) and chemical environmental factors (Chl-a, TOC, TN) were the key factors affecting the changes in the microbial communities in the reservoir water. In addition, a hierarchical partitioning analysis indicated that T, Chl-a, ORP, TOC, pH, and DO accounted for 24.1%, 8.7%, 6.7%, 6.2%, 5.8%, and 5.1% of such changes, respectively. These results are consistent with the ABT (aggregated boosted tree) analysis for the variations in the functional bacterial community, and provide a theoretical basis for the development and application of biotechnology.

## 1. Introduction

In recent years, the water quality in reservoirs and lakes providing drinking water to cities has become an increasingly prominent issue. Algal outbreaks [1], runoff intrusion [2], release of sediments [3], and other types of input pollution [4] have been widely reported. Artificial lakes used as reservoirs usually experience shorter retention times and more intense water level fluctuations than natural lakes. Therefore, different physical, chemical, and biological processes can occur in these aquatic systems [5]. For instance, in subtropical areas, during the late autumn and early winter, reservoirs and lakes experience a natural mixing process. During this mixing, water containing pollutants released from the sediment under long-term anaerobic conditions is incorporated into the whole water column. This causes water pollution in the entire water body. In our research, we studied a pollution issue in Jinpen Reservoir. In this reservoir, a nitrogen concentration of nearly 1.7 mg/L was measured, which is a value exceeding the Chinese standards. Total phosphorus and iron, which exceeded the standards seasonally, also were sometimes a problem for the reservoir.

In order to address pollution in reservoirs, researchers have provided many methods for water quality improvement. For instance, enhanced coagulation to remove dissolved organic carbon was studied, yet it was found useful only for removing large or hydrophobic organic molecules, but not for small or hydrophilic molecules [6]. A reasonable control of water residence time can slow the rate of total nitrogen (TN) increase in water bodies acting as N sinks, but can accelerate its removal from water bodies acting as N sources [7,8]. The use of calcium silicate hydrate to remove phosphorus in water was attempted, yet safety issues arose [9]. Aerobic denitrifying bacteria were found to play a role in water quality improvement in reservoirs, especially for N removal [10]. In summary, physical, chemical, and biological technologies have been used for pollutant removal. However, neither physical nor chemical methods have been able to remove the pollutants completely, despite being high-cost methods [11]. On the other hand, bioremediation—a biological technology entailing the use of microorganisms—has the advantage of not producing secondary pollution or residues while eliminating or reducing the concentrations of hazardous wastes in contaminated sites. 

Microbial communities play a key role in the decomposition of organic matter and the recycling of nutrients in freshwater ecosystems [12]. Differences in the diversity level of microbial composition and community structures among oil reservoirs were evaluated by environmental variation, and the results were useful for the application of indigenous microbial communities [13]. The effects of wet and dry seasons on the bacterial community structure of the Three Gorges Reservoir were researched by Chen [14] using denaturing gradient gel electrophoresis analysis of the PCR-amplified bacterial 16S rRNA gene, and the dominant population was identified. The microbial community has also been analyzed in membrane treatment. It was found that the addition of poly aluminum chloride can improve the performance of the membrane bioreactor by developing different bacterial species and controlling the assailable organic carbon and the associated biofouling on the membranes [15]. Indigenous microbial community variation was observed, which helped to develop and apply more microbial-enhanced oil recovery processes [16]. Indigenous microorganisms in porous concrete may play an important role in pollutant removal from surface water, and *Proteobacteria* were found to dominate in bacterial communities in both planted and unplanted porous concrete systems [17]. In Lake Bourget, the seasonal succession of the main bacterial was observed, and their impact on bacterial structure and dynamics was determined to be likely relevant in similar ecosystems [18]. Microbial diversity and richness in a wastewater treatment plant were found to be negatively associated with elevation, and positively associated with the water temperature to a certain extent [19]. Bacterial abundance was positively correlated with the total phosphorus (TP) concentration in the Qingcaosha Reservoir [20]. Furthermore, significant spatial and temporal changes were observed in the microbial community composition of an estuarine reservoir [21]. The above reports suggest that the microbial community is a key factor in improving water quality. The evolution of microbial population structure can help us better understand the mechanism of water purification.

With regard to in situ water quality purification, mixing and aeration technologies have also been used, and satisfactory results have been achieved [22]. A water-lifting aeration system was installed in the drinking water reservoir which we were investigating, giving us the opportunity to explore the variation in microbial population structure in situ. In a previous study, a satisfactory improvement in water quality was achieved [23]. In this study, we carried out field research in a canyon-shaped drinking water reservoir, the Jinpen Reservoir.

Our research was motivated by the following research questions: (1) At field scale, what is the performance of a water lifting aeration system in terms water quality improvement? (2) How does the microbial community structure vary between the natural state and that resulting from artificial mixing? (3) Which environmental factors can affect the change of microbial communities, and what is the relative influence of the environmental factors?

This study was designed to specifically address these questions. The variation in the microbial community structures in two areas was studied through high-throughput sequencing. Furthermore, principal component analysis (PCA), permutational analysis of variance (PERMANOVA), redundancy analysis (RDA), and ABT were utilized to identify the main factors influencing the microbial community structure evolution. 

## 2. Materials and Methods 

### 2.1. Sampling Sites and Field Work

The Jinpen reservoir (34° 13′ N to 34° 42′ N, 107° 43′ E to 108° 24′ E) is the source of drinking water for the city of Xi’an in Shaanxi Province. It is a large reservoir with a capacity of 2.0 × 10^8^ m^3^, and a mean depth of about 80 m. In the reservoir, 8 water lifting aerators are installed at various locations [23] in order to improve water quality. Two sampling sites, which we termed here “enhanced area” (E) and “control area” (C) were selected, the locations of which are shown in Figure 1. “E” was located in the middle of the artificial mixing area. “C” was located far away (2 km) from the artificial mixing area. Hence, site E and site C were the representatives of the experimental area and comparison area, respectively. In each sampling site, a surface water (0.5 m depth) sample, middle water (40 m depth) sample and bottom water (80 m depth) sample were collected. Each water sample had three parallel samples. When the water temperature difference between surface water and bottom water was less than 1 °C, it was considered that the water was completely mixed.

During the operation of the water-lifting aeration system, water samples were collected every 3 days. Water temperature, dissolved oxygen (DO), oxidation-reduction potential (ORP), turbidity, Chl-a (chlorophyll-a), and pH were examined every 3–5 m in situ by a Hach DS5 multifunctional water quality parameter analyzer. Water quality indices, such as total nitrogen (TN), total phosphorus (TP), and total organic carbon (TOC) were examined in the laboratory, and water samples were collected in triplicate every 10 m. Water samples from the enhanced and control areas were stored in pre-cleaned high-density polyethylene bottles. The samples were immediately cooled and stored at 4 °C until analysis took place.

The objective of mixing aeration was achieved through the operation of a water-lifting aeration system. The operating conditions of the water-lifting aeration system were as follows: Eight water-lifting aerators were running at full load, and each gas supply volume was approximately 50 m^3^/h, 24 h per day, from 28 September 2018 to 29 October 2018.

### 2.2. Physical and Chemical Analysis

Temperature, DO, pH, Chl-a, turbidity, and ORP were determined in situ every 3–5 meters using a Hach multi-probe water quality analyzer (Hydrolab DS5, Loveland, CO, USA). The concentrations of TN, nitrate, ammonia, and nitrite were determined using a SEAL AA3 HR AutoAnalyzer (SEAL, Hamburg, Germany). TOC concentrations were measured with a TOC-L TOC analyzer (Shimadzu, Kyoto, Japan) [24].

### 2.3. High-Throughput Sequencing

A 1-liter water sample was collected, and the water sample was filtered by a 0.22 μm acetate fiber membrane using a vacuum filter pump. The filtered membrane was stored at −22 ℃. DNA extraction and Illumina MiSeq high-throughput sequencing were carried out by Meiji biology Co., Ltd. (Shanghai, China). High-throughput sequencing was mainly used to study the composition of the microbial communities. Using primers 27F (5’-AGAGTTTGATCCTGGCTCAG-3’) and 338R (5’-TGCTGCCTCCCGTAGGAGT-3’) to amplify the V2 region of bacteria in the 16S rRNA gene by PCR [25], PCR amplification was carried out in triplicate using ABI GeneAmp^®^ 9700 PCR System (Saibaiao, Beijing, China) in a total volume of 20 μL PCR reaction mix. All PCR products were separated by 2% gel electrophoresis. The amplicons with sequences shorter than 200 bps and of low quality (quality score < 25) were removed. High-quality sequences were clustered into operational taxonomical units (OTU) at a 97% similarity level and singleton OTUs were eliminated. Meanwhile, the taxonomic classification of effective sequences was determined using the RDP (Ribosomal Database Project) database: http://rdp.cme.msu.edu/. For fair comparison, the size of each sample was normalized to the same sequencing depth by randomly removing the redundant reads. Alpha diversity, beta diversity, and microbial community composition were analyzed based on the OUT results [26]. Based on the OTU clustering analysis results, OTU diversity indices and the depth of sequencing were researched. Based on the taxonomic information, a statistical analysis of the community structure was carried out at each taxonomic level [27]. On the basis of the above analysis, a series of statistical and visual analyses of community structure and phylogeny can be carried out [28].

### 2.4. Data Analysis

The statistical and visual analysis was showed as follows: We tested for mixing and stratification of water effects on community dissimilarity with permutational analysis of variance (PERMANOVA) using the functions adonis in the vegan package with 10^4^ permutations [29]. For each environmental variable, we performed hierarchical partitioning based on Hellinger-transformed rarefied OTU matrix using the rdaenvpart R package [29] and ABT (aggregated boosted tree) analysis [30] to examine the correlation between environmental variables and microbial community composition. 

## 3. Results and Discussion

### 3.1. Effects of Mixing Aeration on Water Quality

The operation of the water-lifting aeration system lasted for about one month from 28 September to 29 October 2018. During this period, the stratification of the reservoir was almost completely destroyed. On 28 September (the first day of operation), the surface water temperature and the bottom water temperature in the enhanced area were 19.8 and 9.9 °C, respectively, for a temperature difference of 9.9 °C, as shown in Appendix A. At the end of the operation (29 October), the water temperature difference had decreased to 3.1 °C. At this time, the average air daily temperature had dropped to 10 °C, and the water temperature had started to decrease rapidly along with the air temperature. The water-lifting aeration system was able to achieve a link between artificial forced mixing and natural mixing. Compared with the enhanced area, the control area was still in a stratification state, and the temperature difference between the bottom water and surface water was 8 °C.

At the same time, variation in DO in the water column was also significant; this indicates that the aeration rate of the water-lifting aeration system was high. As shown in Appendix A, on 28 September, the thickness of the anaerobic zone (DO of 0 mg/L) was almost 9 m; this indicates that the bottom water body had been in a severe anaerobic state for a long time. On 29 October, the DO concentration of the bottom water was 4.2 mg/L, achieving an aerobic condition. With the operation of the water-lifting aeration system, there was an area of significant increase in DO about 6 m above the bottom water, which was due to oxygenation by isothermal layer aeration. In the control area, the DO concentration was still 0 mg/L in the bottom water.

The reduction in TN and TP by the operation of the water-lifting aeration system was mainly due to two mechanisms: First, the release of ammonia nitrogen and phosphorus from sediments was inhibited by increasing DO in the bottom water; second, the continuous mixing effect improved the temperature and DO concentrations in the middle and lower water layers in the reservoir, enhancing the activity of aerobic denitrifying microorganisms in the water, and improving the mixing oxidation conditions of the reservoir, which promoted the oxidation and precipitation of phosphorus in the water. As shown in Figure 2, after operating the water-lifting aeration system, the TN and TP concentrations both obviously decreased. The average concentration of TN decreased from 1.82 mg/L at the start of operation to 0.95 mg/L at the end of operation, which was a reduction of 47.8%. The average concentration of TP decreased from 0.036 to 0.012 mg/L, which was a reduction of 66.7%. In comparison, the TN and TP in the control area decreased by only 14.7% and 18.8%, respectively. Nitrogen removal attracted more and more researchers’ attention. From the perspective of reservoir management, water level control can also help to remove nitrogen [31]. Recently, a novel aerobic denitrifying fungus was reported as having high nitrogen and organic matter removal [32], all of these likely enriched the mechanism of nitrogen removal. 

As shown in Figure 3, TOC concentrations in the enhanced area and control area present the same trend as the TN concentrations. The TOC of the enhanced area decreased from 3.5 to 2.7 mg/L, which was a reduction of 22.9%. In the control area, the TOC decreased by only 12.5%. The same trend was observed in our previous study. As the results showed in other drinking reservoirs, bacterial production played a very important role in dissolved organic matter degradation: dissolved organic matter degradation was high enough to decrease the loads to reservoirs considerably [33].

Based on these results, the stratification in Jinpen Reservoir was nearly destroyed by the running of the water-lifting aeration system; the bottom water layer was oxidized, and DO concentration increased to 4.2 from 0 mg/L. TN, TP, and TOC contents in the water column of the enhanced area were significantly reduced. The running of the water-lifting aeration system not only provided oxygen to the reservoir, but also assisted in improving the water quality. There is no doubt that changes in the water structure and water quality would inevitably affect changes in the water microorganisms [34]. DO and nitrogen were likely the major factors that influenced the microbial communities found in the raw water [35]. Thus, based on these water quality analyses, the microbial community structures in both the enhanced and control areas were explored.

### 3.2. Effects of Mixing Aeration on Microbial Community Structure

Changes in microbial community structure in the enhanced and control areas were explored by using the high-throughput sequencing. As shown in Table 1, a total of 701,702 valid sequences and 14,091 OTUs (97% similarity) were obtained by sequencing. The AEC index varied mainly between 617 and 1205, and the Chao diversity index varied between 616 and 1122. The average coverage of the enhanced area and the control area reached 0.9938 and 0.9940, respectively, indicating that the sequencing results reflect the microbial community structure well [36]. The microbial community diversity and richness in the enhanced area were highly improved over those of the control area by the mixing aeration, especially in the surface water. For example, on 15 October the Chao1 of the surface water was 1004 in the enhance area, and that in the control area was 641. The shannon index of the surface water in the enhanced area on 15 October was 4.32 and that in the control area was 4.04. After running the water-lifting aeration system, the whole reservoir trended to mixing; on 28 November, the values of microbial diversity and richness tended to be consistent. The mixing aeration helped to increase the microbial diversity index. Compared with the Three Gorge Reservoir, less OTUs were obtained in Jinpen Reservoir in the control area [37]. Both results showed that local water quality played an important role in the distribution of bacterial community.

Microbials belonging to seven main categories were found during the operation of the water-lifting aeration system: *Proteobacteria*, *Bacteroidetes*, *Actinobacteria*, *Cyanobacteria*, *Firmicutes*, *Chloroflexi,* and *Verrucomicrobia* (Figure 4 and Figure 5, Appendix A). Among them, *Proteobacteria*, *Bacteroides,* and *Actinomycetes* accounted for 67.52%–78.74% of the total bacterial population. 

During the operation of the water-lifting aeration system, the proportion of *Proteobacteria* in the control area decreased from (44.76 ± 1.75)% to (42.93 ± 1.34)%, whereas in the enhanced area it increased from (43.09 ± 0.96)% to (44.28 ± 1.78)%. The proportion of *Bacteroidetes,* which are well-known degraders of organic matter, decreased from (18.26 ± 0.95)% to (14.45 ± 1.34)% in the control area, while in the enhanced area it decreased from (15.92 ± 1.28)% to (15.02 ± 0.46)%. The variation in *Bacteroidetes* shows that the TOC in the enhanced area degraded faster than that in the control area. Furthermore, the proportion of *Actinomycetes,* which play a crucial ecological role in the recycling of refractory biomaterials and DOM [38], increased in the enhanced area from (11.65 ± 1.75)% to (12.25 ± 0.78)%, while it changed little in the control area. As for *Cyanobacteria,* they were a key group responsible for environmental problems associated with eutrophication processes [39]. In Jinpen Reservoir, *Cyanobacteria* was also a very important monitoring indictor. Research showed that TP and water clarity were identified as the most influential environmental factors correlated with phytoplankton community composition [40]. In our next research, more water quality parameters will be taken into consideration for further analysis on microbials.

*Alphaproteobacteria*, a class of *Proteobacteria*, have been reported to contribute to DOC removal [41]. During the process of mixing aeration, the *Alphaproteobacteria* in the control and enhanced areas showed first a decreasing, and then an increasing trend. In the control area, *Alphaproteobacteria* decreased from (15.27 ± 1.07)% to (14.40 ± 0.73)%, and finally increased to (16.12 ± 0.94)%. That of the enhanced area decreased from (13.74 ± 0.96)% to (13.32 ± 0.87)%, and finally increased to (15.50 ± 1.39)%. The *Betaproteobacteria* in the control system showed a decrease, from (12.30 ± 0.86)% to (11.65 ± 0.06)%. There was an obvious increase in the enhanced area, from (10.17 ± 1.04)% to (12.92 ± 0.76)%. The *Gammaproteobacteria* in the control area decreased first and then increased, from (9.02 ± 0.80)% to (8.08 ± 0.78)% in the beginning, and finally increased to (9.32 ± 0.92)% at the end. There was an obvious increase in the enhanced area, from (9.38 ± 0.55)% at the beginning to (10.55 ± 0.46)%. The *Deltaproteobacteria* in both the control and enhanced area showed a decrease: that in the control area decreased from (6.39 ± 0.68)% to (3.82 ± 0.38)%, and in the enhanced area decreased from (7.65 ± 0.79)% to (3.07 ± 0.69)%. The changes in *Epsilonproteobacteria* in the control and enhanced areas were not significant. This showed that the operation of the aeration system had a greater effect on the enhanced area. Considering the microbial abundance increasing in the enhanced area, as reported in our another research [42], a conclusion might be drawn that aeration and mixing helped improve microbial activity and metabolism of pollutants.

During the process of aeration, as shown in Figure 6, the phylum *Flavobacterioides* in the control area decreased from (7.40 ± 1.16)% to (6.03 ± 0.54)%, and in the enhanced area increased from (5.91 ± 0.26)% to (7.10 ± 0.38)%, and finally to (6.26 ± 0.03)%. The *Sphingomycetes* in the control area decreased from (8.11 ± 0.95)% to (5.70 ± 0.39)%, and in the enhanced area increased from (7.00 ± 1.10)% to (8.12 ± 0.18)%, and finally decreased to (6.49 ± 0.44)%. The microbial community structure in the control and enhanced areas showed different change processes, which shows that the microbial community structure in the water body could be changed by the operation of the water-lifting aeration system. *Proteobacteria* played a very important role in DOC and N cycles, so more attention should be paid to *Proteobacteria*. Considering that bacterial diversity differed depending on the drinking water distribution, *Proteobacteria* could be taken out of the original reservoir and an experiment on its decontamination characteristics should be carried out in the next research.

The changes in the bacteria of the main genera in the water column of the control area and the enhanced area during the operation of the water-lifting aeration system (for proportions of genera >1%) are shown in Figure 7. In the control area, *Cyanobacteria* increased from 6.94% (2018-09-28) to 9.03% (2015-10-28), then decreased to a final value of 4.09% (2018-11-28); *Parcubacteria_norank* increased from 0.65% (2018-09-28) to 1.72% (2018-10-28), then to 5.69% (2018-11-28); *hgcI_clade* decreased from 3.69% (2018-09-28) to 2.48% (2018-09-28). *Flavobacterium* decreased from 2.82% (2018-09-28) to 0.86% (2018-10-28), and then increased to 2.19% (2018-11-28); *Fluviicola* showed a downward trend, from 3.25% (2018-09-28) to 2.58% (2018-10-28), and then decreased to a final value of 1.90% (2018-11-28); *Nocardioides* decreased from 0.43% (2018-09-28) to 0.22% (2018-10-28), and then decreased to 1.17% (2018-11-28); *TM6_norank* rose from 0.00% (2018-09-28) to 1.17% (2018-11-28).

In the enhanced area of the lifting aeration system, *Cyanobacteria* decreased from 6.46% (2018-09-28) to 4.00% (2018-10-15), and then to 5.12% (2018-11-28), and *Parcubacteria_norank* increased from 1.12% (2018-09-28) to 2.46% (2018-10-15), and then to 1.08% (2018-10-28) and finally to 6.36% (2018-11-28) at the end of the operation of the lifting aeration system. *HgcI_clade* increased from 1.54% (2018-09-28) to 2.47% (2018-10-28), and finally to 2.65% (2018-11-28); *Flavobacterium* increased from 1.26% (2018-09-28) to 2.46% (2018-10-15), and then to 2.93% (2018-10-28) at the end of the operation of the water-lift aeration system, and finally to 2.12% (2018-11-28); *Fluicola* remained basically unchanged from 1.94% to 2.46%; *Subgroup_6_norank* increased from 0.98% (2018-09-28) to 1.54% (2018-10-15), *Mitochondria_norank* increased from 0.28% (2018-09-28) to 0.77% (2018-10-15), then to 1.08% (2018-10-28) and finally to 1.06% (2018-11-28). The variation of *Cyanobacteria* showed that water lifting aeration had the function of controlling algae propagation. A new research indicated that temperature control should be considered as a component of water management practices in controlling algae [43].

In conclusion, there were differences in the bacterial changes between the enhanced area and the control area, and with the operation of the water-lift aeration system, the main genera of bacteria varied greatly in time and space.

### 3.3. PCA of Microbial Community

PCA was used to explore the temporal and spatial changes in microbial communities with the operation of the water-lifting aeration system. PCA based on the phylum, class, and genus classification level is shown in Figure 8A–C. 

At the phylum classification level (Figure 8A), PC1 and PC2 account for 60.24% of the total; the cumulative contributions of PC1 and PC2 are 48.37% and 11.86%, respectively. The vertical distribution of microorganisms in different regions was different at the beginning of the experiment. As the experiment progressed, the microbial structure of the enhanced area obviously evolved, and the difference between the two areas remained. The vertical differences within the enhanced area during operation were smaller than those in the control area, which was consistent with the artificial mixing effect of the operation of the water-lifting aeration system. 

At the class level (Figure 8B), PC1 and PC2 account for 50.95% of the total; the cumulative contributions of PC1 and PC2 are 35.52% and 15.43%, respectively. The microbial community structure in the enhanced area and the control area showed obvious spatial and temporal differences. The sample distribution at the same time was more aggregated; and the samples in different areas were obviously separated. 

At the taxonomic level of genera (Figure 8C), PC1 and PC2 account for 23.05%; the cumulative contributions of PC1 and PC2 are 12.56% and 10.49% respectively. The vertical differences in microbial community structure in the enhanced area decreased gradually with the operation of the water-lift aeration system; the samples in different areas and periods were distributed in different locations on the PCA map, and the microbial community in the control and enhanced areas were distributed in different locations. The structure of the microbial community in the two areas became similar one month after the end of the aeration operation. Meanwhile, the water community composition presented significantly difference between enhanced area and control area (F = 2.82, *p* = 0.002 < 0.01) based on PERMANOVA analysis (999 permutations).

Unlike in the Zhoucun Reservoir experiment, the Jinpen Reservoir was a canyon-shaped deep-water reservoir. The operation of the water-lifting aeration system had a weak influence at the periphery. The microbial community structure of the two areas showed obvious differences in the classification levels of the phylum, class, and genera.

### 3.4. Analysis of the Relationship between Environmental Factors and Microbial Communities

As mentioned above, the different microbial communities in the water samples from the enhanced and control areas were discriminated at the phylogenetic and genetic levels. In detail, as shown in Figure 9A, the RDA (redundancy analysis) analysis showed that the first two RDA dimensions (8 parameters, VIF < 20, F = 1.4, *p* = 0.002) could explain 20.13% of microbial community changes, with the contributions of RDA1 and RDA2 being 11.94% and 8.19%, respectively. Physical (temperature, DO) and chemical environmental factors (CHl-a, TOC, TN) are the key factors affecting the changes in the microbial community in the reservoir water. 

As for Figure 9B, based on hierarchical partitioning analysis through rdaenvpart in R, T was found to be the main controlling factor for the variation of bacterial community during the operation of the water-lifting aeration system, which accounted for 24.1% of the variation. The Chl-*a*, ORP, TOC, pH, and DO accounted for 8.7%, 6.7%, 6.2%, 5.8%, and 5.1% of the variation, respectively.

An aggregated boosted tree (ABT) analysis was carried out using R.2.9.1 (gbmplus package) to quantitatively evaluate the relative influence of the water physicochemical factors on bacterial community diversity. For the functional bacteria (Figure 9C) the ABT showed that for C-cycle and S-cycle bacteria DO was the most important driver (33.4%). On the other hand, for CH_4_-cycle, Fe/Mn-cycle, and N-cycle bacteria temperature was the most important driver (26.51%, 28.98%, and 29.91, respectively).

Through the operation of the water-lifting aeration system, the environmental factors of the reservoir water body were changed, thus affecting the changes of water quality.

## 4. Conclusions

The water-lifting aeration system was able to eliminate the existing stratification in the reservoir in the enhanced area. During its operation, TN, TP, and TOC concentrations in the enhanced area experienced a sharper decrease compared with those in the control area: they decreased, respectively, from 1.82 mg/L, 0.036 mg/L, and 3.5 mg/L, to 0.95 mg/L, 0.012 mg/L, and 2.7 mg/L. *Proteobacteria*, *Bacteroides*, and *Actinomycetes* accounted for 67.52–78.74% of the total bacteria evaluated by sequencing. The artificial mixing altered the microbial community structure in the reservoir significantly. For instance, *Cyanobacteria* were transferred to the bottom water. Overall, the results of the analyses showed that artificial mixing and aeration helped to change the microbial community structure. PCA and RDA showed that artificial mixing and aeration greatly changed the microbial community structure in the enhanced area. Microbial community structures in the enhanced area and control area showed an obvious difference.

## Figures and Tables

**Figure 1 ijerph-16-04221-f001:**
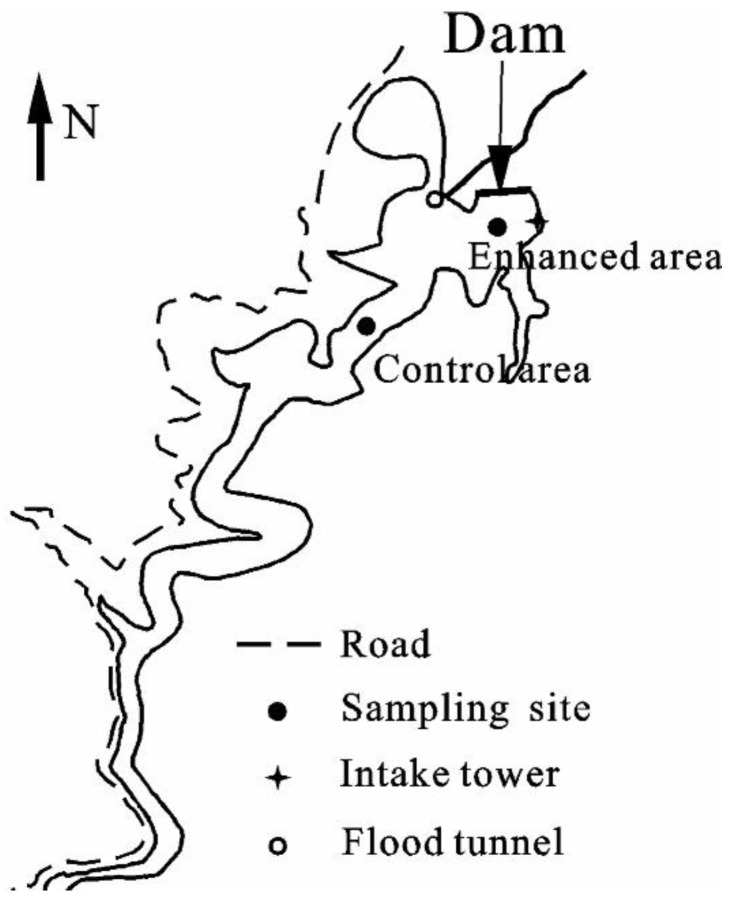
Sampling sites in the enhanced area and control area of Jinpen reservoir.

**Figure 2 ijerph-16-04221-f002:**
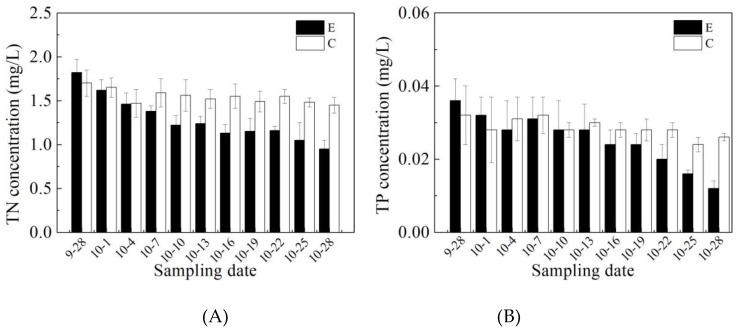
Changes in total nitrogen (TN) (**A**) and total phosphorus (TP) (**B**) concentrations (mean values) in the enhanced area and in the control area during the period of operation of the water-lifting aeration system. In the legends, C stands for “control area” and E for “enhanced area”.

**Figure 3 ijerph-16-04221-f003:**
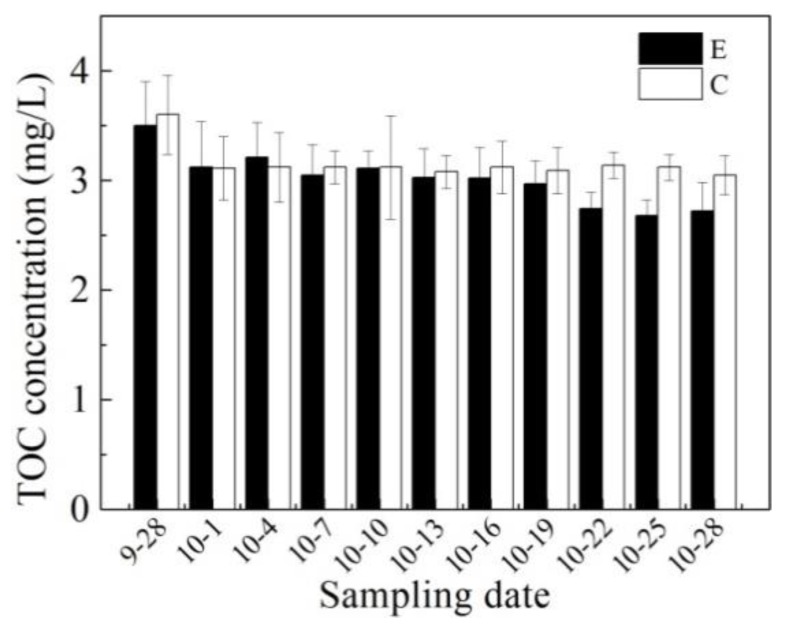
Changes in total organic carbon (TOC) concentration (mean values) in the enhanced area and control area during the running period of the water-lifting aeration system. In the legend, C stands for “control area” and E for “enhanced area”.

**Figure 4 ijerph-16-04221-f004:**
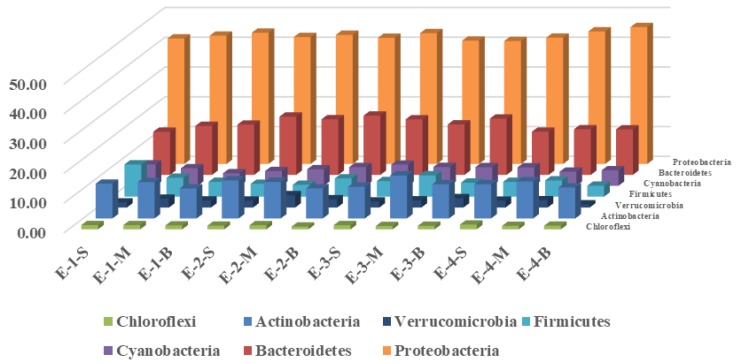
Distribution of main phyla in enhanced areas during the operation of water lifting aerators (The vertical axis represents the percentage of different bacteria; X-Y-Z, area-period-location; X = C and E are control and enhanced areas; Y = 1, 2, 3, 4 refer to 2018-09-28, 2018-10-15, 2018-10-29, 2018-11-28; Z = S, M, B, are surface, medium, and bottom water layer).

**Figure 5 ijerph-16-04221-f005:**
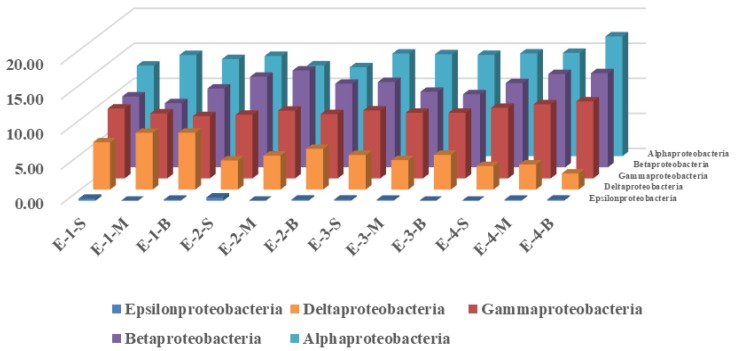
Changes in the proportions of *Proteobacteria* in the enhanced area during the operation of the water-lift aeration system. Labeling is consistent with that of Figure 4.

**Figure 6 ijerph-16-04221-f006:**
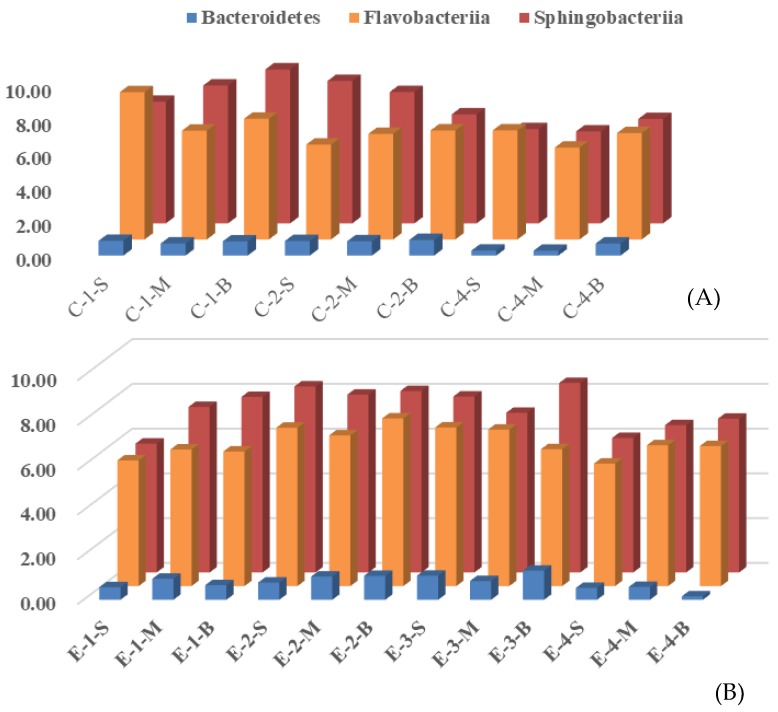
Changes in *Bacteroides* in the reservoir during the operation of the water-lifting aeration system: (**A**) change in the control area; (**B**) change in the enhanced area. Labeling is consistent with that of Figure 4.

**Figure 7 ijerph-16-04221-f007:**
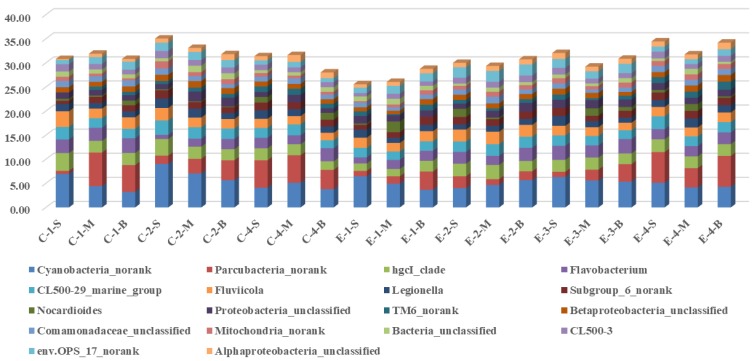
Distribution of the main genera during the operation of the water-lifting aeration system (>1%). Labeling is consistent with that of Figure 4.

**Figure 8 ijerph-16-04221-f008:**
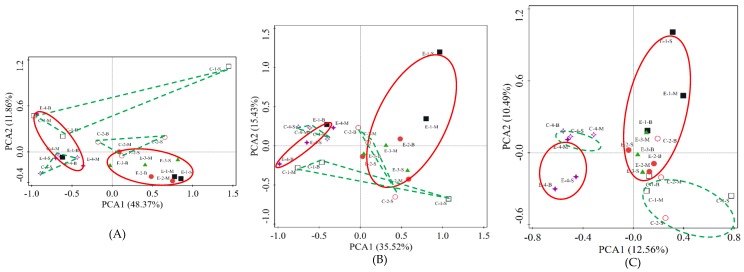
Principal component analysis (PCA) of water samples in the enhanced and control areas: (**A**) at phylum level; (**B**) at class level; (**C**) at genus level.

**Figure 9 ijerph-16-04221-f009:**
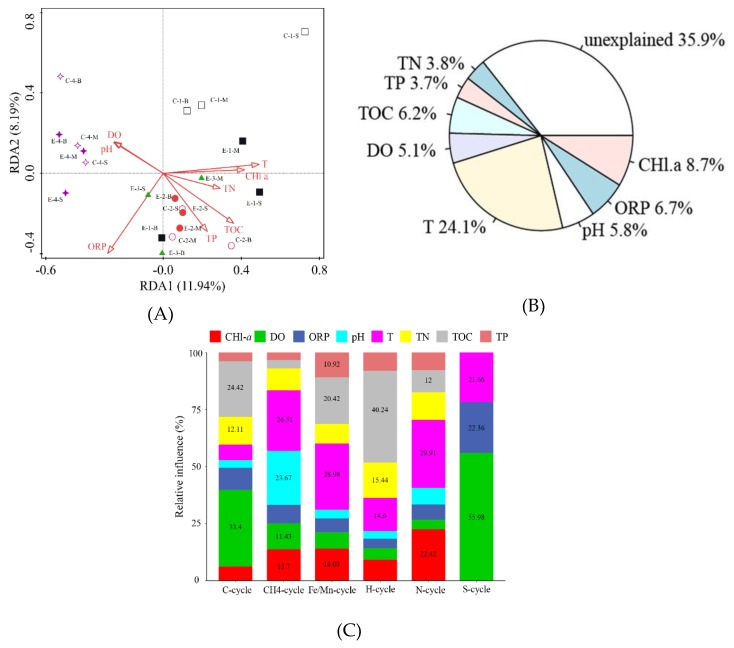
Relationship between environmental factors and microbial communities in Jinpen Reservoir: (**A**) redundancy analysis (RDA); (**B**) hierarchical partitioning analysis; (**C**) aggregated boosted tree (ABT) for functional bacteria.

**Table 1 ijerph-16-04221-t001:** Spatial and temporal distribution of the microbial community diversity and richness estimators in the enhanced and control areas.

Sampling Time	Site (Water Depth)	Enhanced Area							Control Area						
		Reads	0.97 Level						Reads	0.97 Level					
			OTUs	Diversity		Coverage	Richness			OTUs	Diversity		Coverage	Richness	
				ACE	Chao1		Shannon	Simpson			ACE	Chao1		Shannon	Simpson
09–28	0.5 m	24,639	712	931	923	0.9916	4.55	0.0275	42,439	461	617	616	0.9968	3.59	0.0566
	45 m	39,804	852	1034	1021	0.9951	4.85	0.0198	31,783	702	948	934	0.9932	4.09	0.0473
	90 m	40,082	765	1019	1029	0.9943	4.38	0.0313	29,379	717	1205	1065	0.9914	4.15	0.0405
10–15	0.5 m	27,963	650	1105	1004	0.9920	4.32	0.029	27,036	465	600	641	0.9952	4.04	0.0441
	45 m	29,817	579	740	780	0.9947	4.22	0.0338	33,172	595	758	773	0.9953	4.23	0.0379
	90 m	30,418	654	858	877	0.9938	4.45	0.0259	39,454	654	820	839	0.9958	4.38	0.0333
10–29	0.5 m	33,431	649	953	843	0.9946	4.47	0.0256	--	--	--	--	--	--	--
	45 m	31,302	715	960	981	0.9930	4.47	0.0261	--	--	--	--	--	--	--
	90 m	38,056	768	1007	1018	0.9942	4.33	0.0397	--	--	--	--	--	--	--
11–28	0.5 m	19,696	566	905	810	0.9905	4.15	0.0402	26,986	685	918	908	0.9924	4.41	0.0327
	45 m	46,082	699	1045	952	0.9957	4.34	0.0305	34,268	702	914	947	0.9941	4.37	0.034
	90 m	45,379	655	1041	882	0.9955	3.95	0.0502	30,516	846	1111	1122	0.9918	4.91	0.0171

OTUs: operational taxonomical units; ACE: Abundance-based Coverage Estimator.

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
