# Peer review of "Field Research on Mixing Aeration in a Drinking Water Reservoir: Performance and Microbial Community Structure"

_ijerph, 2019, doi:10.3390/ijerph16214221_

Round 1

Reviewer 1 Report

This study describes the effects of a water-lifting aeration system on the aquatic bacterial community composition in a Chinese lake. The patterns of bacterial diversity were analyzed by the MiSeq amplicon sequencing approach and linked to some major physical-chemical characteristics of the water samples through explorative multivariate statistics (PCA, RDA).

In comparison to a control station, the water aeration was effective in removing excess nutrients (TN, TP, and TOC) and altered the bacterial community profiles.

Leaving aside the numerous language inconsistencies, my major concerns are related to the study rationale, aims, and text structure. The paper is not likely driven by a clear research question and, in my opinion, most of the manuscript sections are flawed. Some major issues that raised my attention are detailed here below:

1- The study aims are based exclusively on local descriptive issues (L99-104). A novel and possibly attractive research hypothesis should be clearly stated at the end of the introduction. It is rather trivial that enhanced aeration promoting nutrient removal could consequently affect the microbial community structuring. As presented, it is hard to see the original contribution of this study to the current knowledge.

2- A disproportionate importance is given to the target sites, and results appear of local relevance only. The sampling effort is limited to a single sampling station (+ control). I perfectly understand the work done to elaborate and prepare data for publication, but the number of analyzed samples is not likely sufficient to “explore the temporal and spatial changes of microbial communities” and the potential links with water quality patterns.

3- The water quality was defined only according to a limited set of major parameters (see L131-134). Thus, it is not possible to understand which of and to what extent the pollution level, along with the (myriad) of disregarded environmental factors, could have contributed to the microbial community structuring.

4- The methods lack sufficient details for reproducibility. For example, the 16S data elaboration and statistical analyses have to be fully described to assess the suitability of the applied tests. Does ANOVA meet the assumptions of data normality and homoscedasticity? If not, a the non-parametric analogous test should be applied. How were zero-values considered in the PCA and RDA?

5- Results and discussion are mixed in a single section. This could be acceptable for the journal, but the reported results appear only partly elaborated and the points of discussion are very few. My suggestion is to divide results and discussion in two distinct sections, thus presenting concisely the main results and focusing on relevant, updated, and literature-supported discussion points.

6 - the English language is far below the standard required for international publications. Despite I am not a native English speaker myself, I would recommend asking the help of a mother-tongue colleague or a proofreading service to correct the numerous language inconsistencies found all over the text.

Reviewer 2 Report

The research has its merits and it is very interesting. The water crisis it is affecting everyone in different degrees; so research like this gives the scientific community a sense of what can be made to improve our reservoirs.
1. The introduction can be improved; more references should be collected, directing the reader towards the scope of the problem at hand.
2. Figure #1 should be clearer.
3.  On line #147 On the basis of the above analysis, a series of statistical and visual analyses of community structure and phylogeny can be carried out Explain those analysis.
4. On Results and Discussion it is not fully explained the effects of the fluctuations of the parameters on the microbiome of the site. For example:
 1) How a drop on water temperature can affect the microbes present?
 2) Is the sampling month is representative of the entire year? Explain
5. Table #1 should be explained in more detail
 1) How depth affect the microbiome? and how the control and experimental area compare?
6.  In the conclusion, how the changes in microbial structure change the water quality? Why is this important?
Overall, the research is very commendable and I recommend publication with this minor changes.

Reviewer 3 Report

This paper aims to investigate the microbial community structures and pollutant removal performance in a drinking water reservoir under mixing aeration condition. Water quality problems of reservoirs has been paid great attention all over the world. Compare to physical, chemical technologies, such as coagulation, biological technology became a good choice because of no secondary pollution. The presented results can provide useful reference to both researchers in lab and the industries. However, this study is not well presented in the current version of manuscript, especially in terms of description of the abstract, methods and conclusions. Besides, I strongly encourage the authors to have this manuscript edited by an English-language service or by a native English speaker. A major revision is required.

1.The water-lifting aeration system was operated in autumn for one month, why did the authors choose this season. Temperature, rain runoff pollution and other environmental factors have significant effect on the bacterial structure. I do not think only one month operation in autumn could comprehensively delineate the overall situation.

2.Aeration system is efficient for algal control, the trend of algal abundance should be considered.

3.Introduction expressed too much redundant information. Please shorten the introduction section, especially the third paragraph.

4.The main novelty of this study should be clarified in the introduction.

5.How many liters of water sample for microbial community analysis was collected? And How to concentrate the water? How to extract the DNA? The methods and material section should be carefully revised.

6.The legends of Fig2 and Fig 3, what does C and E represent? Please double check all figure legends.

7.Line 239:” location;X=C 238 and E mean control and enhanced area”. Which sample is the control area? I cannot find it.

8.Fig 9 is not necessary, RDA results showed in table 2 is enough.

9.Please check and double check all references.

10.Please revise the conclusion to emphasize your main results.

11.Error bar should be added in the figures of the supplemental materials.

Round 2

Reviewer 1 Report

The revised version of this study provides further information and data elaboration, but the overall structure and readability is still rather fuzzy.

I appreciate the work done to better contextualize the study rationale and aims, but it is hard to understand how the presented data could contribute to address the research questions at L93-96:

- “How is the performance of water lifting aeration system in water quality improvement?”

How and what are “performance” meant for?

- “Then how is the microbial community structure variation between the artificial mixing and natural state?”

This is what was done and not what was entailed to be tested. It is obvious that the microbial community structure change between the artificial mixing and natural state.

- “Which environmental factors can affect the change of microbial communities and what’s the relative influence of the environmental factors?”

I believe that “the relative influence of the environmental factors” largely depends on the (limited) number of water parameters herein presented. The definition and resolution of water quality is too low to derive any speculations out of a multivariate approach.

- The result are not yet discussed in the context of the current literature (e.g., there are only 3 reference "local" studies cited in the result and discussion).

- I see again too many language and punctuation mistakes, but this will be an editorial issue.

Author Response

Comments and Suggestions for Authors

The revised version of this study provides further information and data elaboration, but the overall structure and readability is still rather fuzzy. I appreciate the work done to better contextualize the study rationale and aims, but it is hard to understand how the presented data could contribute to address the research questions at L93-96:

- “How is the performance of water lifting aeration system in water quality improvement?” How and what are “performance” meant for?

Reply: Thank you for your comments.

  Here “performance” means results of water quality improvement. For example, after artificial intervention, in the enhanced area, the DO concentration in the bottom water increased to 4.2 mg/L, the temperature difference between the bottom water and surface water decreased to 3.1 ℃. And total nitrogen, total phosphorus, total organic carbon concentrations were reduced by 47.8%, 66.7%, and 22.9%, respectively. These were the meaning of “performance”. Compared with the control area, the artificial mixing helped to remove some pollutants (TN, TP, TOC) in the enhanced area.

- “Then how is the microbial community structure variation between the artificial mixing and natural state?”

This is what was done and not what was entailed to be tested. It is obvious that the microbial community structure change between the artificial mixing and natural state.

 Reply: Thank you for your comments.

  The purpose of artificial mixing is to improve water quality. So enhanced area was the experimental area, which was in the middle of the artificial mixing area. Control area was a contrast area, which was far away from artificial mixing area. Then the difference was studied between these two areas.

  As shown in the Figure-B below, T was the largest influence factor for the variation of bacterial community during the operation of WLAs, which accounted for 24.1% of the whole variation. The Chl-a, ORP, TOC, pH, and DO accounted for 8.7%, 6.7%, 6.2%, 5.8% and 5.1%, respectively. In the enhanced area, the temperature difference nearly disappeared, and that in the control area still existed. The statistical results were consistent with the experimental results.

    So we can see that, through changing water temperature, dissolved oxygen and other parameters, the aeration system affected the structure of microbial community, and ultimately reduced the concentrations of nutrients in the water.

- “Which environmental factors can affect the change of microbial communities and what’s the relative influence of the environmental factors?”

I believe that “the relative influence of the environmental factors” largely depends on the (limited) number of water parameters herein presented. The definition and resolution of water quality is too low to derive any speculations out of a multivariate approach.

 Reply: Thank you for your comments.

  Indeed, the water quality parameters that we used in this research were a little less. More water quality indicators should be studied to further illustrate our speculations. Of course, your suggestions gave us guidance that more environmental factors would be considered in our future research. According to our monitoring, TN and organic matter are the exceeded standard indexes compared with 《Environmental Quality Standards for Surface Water (GB 3838-2002)》. So in this research only the key indexes. In the RDA and PCA analysis, only limited water quality indicators were discussed.

- The result are not yet discussed in the context of the current literature (e.g., there are only 3 reference "local" studies cited in the result and discussion).

  Reply: Thank you for your comments.

We added some discussion in the manuscript, and marked red in the manuscript.

The contents are as following:

Nitrogrn removal attracted more and more researchers’ attention. From the perspective of reservoir management, water level control can also help to remove nitrogen [31]. Newly, a novel aerobic denitrifying fungus was reported of high ntrigen and organic matter removal [32], all of these enriched the mechanism of nitrogen removal.

As the results showed in other drinking reservoirs, bacterial production played a very important role in dissolved organic matter degradation: dissolved organic matter degradation was high enough to decrease the loads to reservoirs considerably [33].

Compared with the Three Gorge Reservoir, less OTUs were obtained in Jinpen Reservoir in the control area [37]. Both results showed that local water quality played an important role in the distribution of bacterial community

As for Cyanobacteria, they were a key group responsible for environmental problems associated with eutrophication processes [39]. In Jinpen Reservoir, Cyanobacteria was also a very important mornitoring indictor. Research showed that TP and water clarity were identified as the most influential environmental factors correlated with phytoplankton community composition [40]. In our next research, more water quality parameters would be taken into consideration for further analysis on microbials.

Considering that bacterial diversity differed depending on the drinking water distribution, Proteobacteria could be taken out from the original reservoir and an experiment on its decontamination characteristics should be carried out in the next research.

The variation of Cyanobacteria showed that water lifting aeration had the function of controlling algae propagation. A new research indicated that temperature control should be considered as a component of water management practices in controlling algae [43].

Francisco J.; David H.; et al. Effective depth controls the nitrate removal rates in a water supply reservoir with a high nitrate load. Sci. Total Environ. 2019, 673: 44-53. Haihan Z.; Zhenfang Z.; Pengliang K.; et al. Biological nitrogen removal and metabolic characteristics of a novel aerobic denitrifying fungus, Hanseniaspora uvarum, strain KPL108. Bioresour. Technol. 2018, 267: 569-577. Norbert K.; Marieke R.; et al. Bacterial production and their role in the removal of dissolved organic matter from tributaries of drinking water reservoirs. Sci. Total Environ. 2016, 548-549: 51-59. Park Y.; Cho H.; Yu J.;, Min B.; Hong S.K.; Kim B.G.; Lee T. Response of microbial community structure to pre-acclimation strategies in microbial fuel cells for domestic wastewater treatment. Bioresource Technol. 2017, 233: 176-183. 37. Aping N.; Liyan S.; Yanghui X.; et al. Impact of water quality on the microbial diversity in the surface water along the Three Gorge Reservoir (TGR), China. Environ. Safe. 2019, 181: 412-418. Hassan S.S.; Anjum K.; Abbas S.Q.; Akhter N.; Shagufta B.I.; Shah S.A.; Tasneem U. Emerging biopharmaceuticals from marine actinobacteria. Environ. Toxic. Phar. 2016, 49: 34. Tatenda D.; Wasserman R.J. Cyanobacteria dynamics in a small tropical reservoir: Understanding spatio-temporal variability and influence of environmental variables. Sci. Total Environ. 2018, 643: 835-841. Beaver J. R.; Tausz C.E.; Scotese K.C.; et al. Environmental factors influencing the quantitative distribution of microcystin and common potentially toxigenic cyanobacteria in U.S. lakes and reservoirs. Harmful Algae, 2018, 78: 118-128. He W.; Luo J.; Xing L.; et al. Effects of temperature-control curtain on algae biomass and dissolved oxygen in a large stratified reservoir: Sanbanxi Reservoir case study. J. Environ. Manage. 2019, 248: 109250.

- I see again too many language and punctuation mistakes, but this will be an editorial issue.

 Reply: Thank you for your comments.

We contacted Language Editors for further language verification, again.
